# Liver Injury in COVID-19 Patients with Drugs as Causatives: A Systematic Review of 996 DILI Cases Published 2020/2021 Based on RUCAM as Causality Assessment Method

**DOI:** 10.3390/ijms23094828

**Published:** 2022-04-27

**Authors:** Rolf Teschke, Nahum Méndez-Sánchez, Axel Eickhoff

**Affiliations:** 1Department of Internal Medicine II, Division of Gastroenterology and Hepatology, Klinikum Hanau, Academic Teaching Hospital of the Medical Faculty, Goethe University Frankfurt, 63450 Hanau, Germany; med2@klinikum-hanau.de; 2Liver Research Unit, Medica Sur Clinic & Foundation, Mexico City 14050, Mexico; nmendez@medicasur.org.mx; 3Faculty of Medicine, National Autonomous University of Mexico, Mexico City 04510, Mexico

**Keywords:** COVID-19, DILI, drug-induced liver injury, Roussel Uclaf Causality Assessment Method, RUCAM

## Abstract

Patients with coronavirus disease 19 (COVID-19) commonly show abnormalities of liver tests (LTs) of undetermined cause. Considering drugs as tentative culprits, the current systematic review searched for published COVID-19 cases with suspected drug-induced liver injury (DILI) and established diagnosis using the diagnostic algorithm of RUCAM (Roussel Uclaf Causality Assessment Method). Data worldwide on DILI cases assessed by RUCAM in COVID-19 patients were sparse. A total of 6/200 reports with initially suspected 996 DILI cases in COVID-19 patients and using all RUCAM-based DILI cases allowed for a clear description of clinical features of RUCAM-based DILI cases among COVID-19 patients: (1) The updated RUCAM published in 2016 was equally often used as the original RUCAM of 1993, with both identifying DILI and other liver diseases as confounders; (2) RUCAM also worked well in patients treated with up to 18 drugs and provided for most DILI cases a probable or highly probable causality level for drugs; (3) DILI was preferentially caused by antiviral drugs given empirically due to their known therapeutic efficacy in other virus infections; (4) hepatocellular injury was more often reported than cholestatic or mixed injury; (5) maximum LT values were found for alanine aminotransferase (ALT) 1.541 U/L and aspartate aminotransferase (AST) 1.076 U/L; (6) the ALT/AST ratio was variable and ranged from 0.4 to 1.4; (7) the mean or median age of the COVID-19 patients with DILI ranged from 54.3 to 56 years; (8) the ratio of males to females was 1.8–3.4:1; (9) outcome was favorable for most patients, likely due to careful selection of the drugs and quick cessation of drug treatment with emerging DILI, but it was fatal in 19 patients; (10) countries reporting RUCAM-based DILI cases in COVID-19 patients included China, India, Japan, Montenegro, and Spain; (11) robust estimation of the percentage contribution of RUCAM-based DILI for the increased LTs in COVID-19 patients is outside of the current scope. In conclusion, RUCAM-based DILI with its clinical characteristics in COVID-19 patients and its classification as a confounding variable is now well defined, requiring a new correct description of COVID-19 features by removing DILI characteristics as confounders.

## 1. Introduction

Coronavirus disease 19 (COVID-19) infections are commonly viewed as systemic diseases as shown already in initial comprehensive studies from China published in 2020 [1,2], based on 41 cases [1] and 1099 cases [2]. Among the most involved organs were the lungs causing acute respiratory syndrome [1,2] with a need for invasive mechanical ventilation [1], kidneys leading to acute renal injury requiring continuous kidney replacement therapy [1], and heart resulting in acute cardiac injury [1,2]. As a consequence, COVID-19 patients may present at various extent clinical features such as fever [1,2], chills [2], septic shock [2], disseminated coagulopathy [2], rhabdomyolysis [2], myalgia [1], arthralgia [2], fatigue [1], conjunctival and nasal congestion [2], sore throat [2], cough [1,2], dyspnea [1,2], sputum production [1], hemoptysis [1,2], nausea [2], vomiting [2], and diarrhea [1,2]. As expected and presented in detail, laboratory test abnormalities are common features among COVID-19 patients, with abnormal liver tests (LTs) such as elevated serum activities of alanine aminotransferase (ALT) and aspartate aminotransferase (AST), but their increases were causally not traced back to individual drugs because, in most cases, only global drug groups such as antibiotic, antiviral, or antifungal drugs were mentioned, and causality assessment was lacking [1,2].

This systematic review focuses on published reports on conventional drugs and their role in causing liver injury among COVID-19 patients, a special study cohort characterized by multiple morbidities and multiple medications. Of special interest were patients under drug therapy, who showed increased LTs or experienced suspected drug-induced liver injury (DILI) in temporal association with the use of drugs, for which a causal relationship was also reported using the worldwide applied RUCAM (Roussel Uclaf Causality Assessment Method). The final aim was to define the characteristic features of DILI in COVID-19 patients.

## 2. Literature Search Strategy and Selection Criteria

This systemic review was based on a systematic search on published reports using Google and the PubMed database with the following keywords: COVID-19 AND liver test abnormalities AND drugs, AND DILI, AND RUCAM. The search returned a large number of publications with great variability depending on the used term combination. The literature search via Google provided 3000 hits using the search term combination of COVID-19, liver test abnormalities, drug, DILI, and RUCAM, whereas 216,000,000 hits were achieved by COVID-19 and liver test abnormalities, and 35,700 hits were achieved by COVID-19, DILI, and RUCAM. Lastly, 21,200 hits were presented by COVID-19 and RUCAM as search terms. The first 50 publications derived through Google from the four groups of combined terms were checked for their suitability to be included in this study and provided the primary base for further analysis. The first search was performed on 30 November 2021 and was then updated on 31 December 2021. Publications were complemented by the large private archive of the authors. There was no limitation by language, year of publication, or study design. The study eligibility was defined by identifying RUCAM-based DILI cases due to drugs used to treat patients with COVID-19. The study selection was performed by two independent reviewers in three sequential stages—title, abstract, and full-text readings. A third reviewer resolved any disagreements. The following variables were analyzed: drug, patient characteristics, assessment of liver test abnormalities, and DILI diagnosis criteria using RUCAM.

The search returned a selection of 200 articles. After excluding 194 reports as duplicate articles and reports not covering the complete spectrum of used RUCAM in COVID-19 patients with suspected DILI, six available abstracts and full texts were assessed, as summarized in the flow diagram of the literature search and analysis process (Figure 1).

## 3. Results and Discussion

### 3.1. Frequency of Abnormal Liver Tests in COVID-19 Patients

Consensus exists that increased LTs are found in a portion of patients with COVID-19 infections as evidenced by reports published in 2020 [1,2,3,4,5,6,7,8,9,10,11,12,13,14,15,16,17,18,19,20,21,22,23,24,25,26,27] and 2021 [28,29,30,31,32,33,34,35,36,37,38,39,40,41,42,43,44,45,46,47,48,49,50,51,52,53,54,55,56,57,58,59,60,61,62]. In more detail, the frequency of abnormal LTs in patients with COVID-19 infections varied substantially among the published reports, ranging from 4.8% to as much as 78% [63] or 76.3% [5]. The broad variability is best explained by the inhomogeneity of the study cohorts and is likely due to differences in reporting countries, timepoint of assessment whether at hospital admission or during hospital stay, differences in disease severity, confounding variables, or different definitions of increased LTs comprising all patients with any serum ALT and AST value above the upper limit of normal (ULN). By this approach, cases were included with small increases in ALT and AST values such as 1–5 times the ULN, as well as patients with real liver injury as evidenced by values more than five times the ULN. LT denotes parameters such as ALT, AST, or alkaline phosphatase (ALP) as opposed to liver function tests (LFTs), which require unconjugated bilirubin as an additional parameter. In some publications, LTs were not clearly differentiated from LFTs, making comparisons of results and conclusions more difficult.

### 3.2. Tentative Causes of Liver Injury

The causes of increased LTs with serum activities of ALT and AST as parameters of interest in COVID-19 patients are still under scientific discussion, as briefly summarized [3,4,5,63,64,65,66,67,68,69,70,71]. Firstly, they may have an extrahepatic origin as organs other than the liver such as the heart and muscles contain these enzymes released into the bloodstream upon external injurious attacks of the organs caused, for instance, by viruses including COVID-19 [5,42] or by bacteria in the context of systemic sepsis [5,52,64,65] or shock [5,64,65,66,67]. Secondly, extrahepatic conditions may reduce the oxygen supply for the liver and cause hepatic hypoxia [3,64,66,67] via mechanisms such as reduced respiration through the impaired respiratory center in the medulla oblongata and the pons due to a cerebral insult [53], by pneumonia and pulmonary embolism in line with acute respiratory distress syndrome (ARDS) [42,53,64,65,68,69,70], or by cardiac problems triggering liver congestion and hypoxia [43,53,66]. Thirdly, they may be caused by the viruses through their hepatotropism [4] and direct cytopathic effect via the angiotensin-converting enzyme 2 (ACE2) receptor due to the cytokine storm [69], entering the hepatocytes or nonparenchymal cells of the liver [3,5] and leading to hepatocellular or cholestatic injury [5] and endothelial injury with thrombosis of liver vessels and aggravating hepatic hypoxia [3]. Fourthly, they may be related to preexisting liver diseases [3,42,53,65,67,68,69] and possibly triggered by the COVID-19 infection [3,53]. Fifthly, they may be seen in the context of concomitant acute virus infection such as hepatitis A–E [5]. Lastly, they may causally be related to the use of herbal medicines including herbal traditional Chinese medicines (TCMs) [5,36,71] or conventional drugs [3,5,42,43,53,63,64,65,66,67,69].

### 3.3. Drug Use in COVID-19 Patients with Abnormal Liver Tests

The number of publications on clinical characteristics of patients infected by COVID-19 is overwhelming, with 110,000,000 results obtained via PubMed and 455,000,000 results received via Google on 2 January 2022. Many of these publications presented data on drugs used for the treatment of COVID-19 patients, who also showed increased LTs; a selection of these reports is listed (Table 1). Listed are only individual drugs rather than drug groups such as antibiotics, antivirals, antifungals, nonsteroidal anti-inflammatory drugs (NSAIDs), or herbal medicines including traditional Chinese medicines (TCMs). The analysis of the listed reports (Table 1) led to the following conclusions: (1) most reports followed a retrospective study design with all its limitations such as incomplete data and lacking appropriate exclusion of alternative causes; (2) the study cohorts were inhomogeneous in contrast to the homogeneity in prospective studies; (3) in some studies, data at admission and during the hospital stay were mixed; (4) occasionally, data obtained with herbal medicines including herbal TCMs were not analyzed separately from those obtained with conventional drugs; (5) little attention was paid to the causal relationship between drug use in general or individual drugs and increased LTs or DILI, an issue requiring further discussion.

### 3.4. Missing Causality Assessment for Suspected Drugs in COVID-19 Patients

It is obvious that many publications on COVID-19 patients described in detail their treatment with drugs and increased LTs in temporal association with the use of drugs, but they failed to evaluate the causality for drugs in these patients (Table 1). As it currently stands, the question remains unanswered whether COVID-9 patients under a drug treatment may or may not have a real DILI disease. This failure certainly obscures and invalidates the published clinical features as they comprise the effects of two potential injurious compounds as confounders, COVID-19 virus and drugs if they were applied to treat the ongoing virus infection or preexisting liver diseases. In most COVID-19 reports, a robust causality assessment for the used drugs was missing, leaving their role for the observed LT abnormalities unanswered (Table 1). In a special cohort of a single study, however, abnormal LTs were found in patients who were not treated by antiviral drugs [68], suggesting that COVID-19 viruses per se may at least partially be responsible for the LT abnormalities in COVID-19 patients.

### 3.5. Reports Recommending the Use of RUCAM

Stimulating are the 14 reports (Table 1) that used or at least recommended RUCAM [72,73] as a valuable approach to assess causality for drugs in suspected DILI, with more reports (*n* = 11) published in 2021 than the three publications in 2020 (Table 1), reminding us to use RUCAM best in its updated version published in 2016 [73] rather than its original version published in 1993 [72]. In fact, the two RUCAM versions were used in six COVID-19 reports with suspected DILI cases (Table 1) [16,32,34,38,40,60], as described in more detail (Table 2). Of note, RUCAM-based case evaluation was conducted by the quoted authors and not by us as case details were not available to us but confined to the authors of their published work. The use of RUCAM allowed them to exclude alternative causes to be differentiated from DILI.

### 3.6. Verified Drugs in 393 RUCAM Based DILI Cases

Using RUCAM to assess causality in suspected DILI cases, highly appreciated are the efforts of six groups to shed more light on the role of drugs in the increased LTs and suspected DILI commonly found in 393 COVID-19 patients under a drug therapy (Table 2) [16,32,34,38,40,60]. They verified for the first time the existence of DILI in COVID-19 patients by means of probable or highly probable RUCAM-based causality. In general, the six publications discussed in Table 2 were a suitable and valuable basis for the published conclusions. It is encouraging to see that at least a few groups realized that RUCAM should be used for suspected DILI cases to verify causality for certain drugs (Table 1 and Table 2). With the exemption of a single report referencing TCZ [16], perfect was the observation that, in the remaining 5/6 reports (Table 2) [32,34,38,40,60], help was not sought from the LiverTox database, which provides poor-quality data of DILI cases due to the lack of a robust causality assessment causing heavy scientific discussions [74,75,76,77,78].

Excellent was the quality of reports of single cases or with a few COVID-19 patients, because details were perfectly presented with a high RUCAM-based causality grading likely achieved by completeness of case data collected during the hospital stay (Table 2). Less perfect were retrospective studies because they provided a high number of DILI cases with a possible causality grading for individual drugs, conditions due to incomplete case datasets that inevitably reduce the total scores of RUCAM. Occasionally, study cohorts combined DILI cases of possible causality gradings with those of probable or higher causality gradings, conditions that partially obscured the results. Rarely, cases with ALT values <5 times the ULN were included that may confound part of the results. Authors had obviously no problem using RUCAM correctly even if multiple drugs were used in the same COVID-19 patient, thereby adding to the reputation RUCAM received worldwide [79].

Overall, it has to be acknowledged that evaluation of DILI in COVID-19 patients is a particular clinical challenge due to possible interacting and confounding factors, and some of the obtained data may need a more cautious interpretation by the publishing authors. Although RUCAM commonly takes care of many alternative causes possibly confounding the diagnosis of DILI [73], problems may theoretically occur but remain speculative in patients with MAFLD (metabolic-associated fatty liver disease), suspected in a few obese COVID-19 patents but not mentioned in the published reports [16,32,34,38,40,60]. RUCAM can help identify a suspected drug and stop its use to prevent further health risks. Considering the complexity of drugs and DILI in the complicated COVID-19 disease, RUCAM certainly cannot solve all upcoming clinical DILI issues, which may need the intellectual input of the physicians. Poor quality of published RUCAM-based DILI cases is rare and commonly the result of disputable qualification of the assessing scientist but cannot be attributed to the validated RUCAM itself, approved by international DILI experts [72] and worldwide in use [79]. Lastly, background noise on RUCAM elements is limited, and improvement attempts were published in several electronic RUCAM-based versions; all were deeply modified but none of the new methods have been validated, preventing common use.

### 3.7. Clinical Features of RUCAM Based DILI in COVID-19 Patients

Out of initial 200 reports derived from the PubMed database, six reports provided 996 cases, from which overall 393 cases of RUCAM-based DILI in COVID-19 patients with abnormal LTs were found suitable for further analysis (Figure 1). Cases of half of the six studies [16,38,40] were assessed for causality using the original RUCAM [72], whereas the updated RUCAM [73] was applied in the other half [32,34,60]. This enabled a clear description of clinical features of DILI in COVID-19 patients, classifying DILI also as a confounding variable in the COVID-19 setting (Table 3). Key elements of DILI are described in condensed form: (1) DILI in the COVID-19 study cohort was preferentially caused by antiviral drugs given empirically in face of their proven efficiency in infection diseases caused by a variety of viruses; (2) the updated RUCAM worked well even in patients with up to 18 co-medicated drugs, providing clear individual causality gradings for each drug used; (3) according to RUCAM criteria using ALT and ALP as diagnostic parameters, hepatocellular injury was more often found than cholestatic or mixed injury; (4) maximum LT values were published for alanine aminotransferase (ALT) 1.541 U/L and aspartate aminotransferase (AST) 1.076 U/L; (5) the ALT/AST ratio was variable and ranged from 0.4 to 1.4; (6) the age of the COVID-19 patients with DILI ranged from 54.3 to 56 years; (7) the ratio of males to females was 1.8–3.4:1; (8) outcome was favorable for most patients, likely due to careful selection of the drugs and quick cessation of drug treatment if DILI developed, but clinical course was related to fatal disease in one patient; (9) countries reporting RUCAM-based DILI cases in COVID-19 patients included China, India, Japan, Montenegro, and Spain; (10) the currently analyzed six reports were not designed to determine the quantitative contribution of DILI and the COVID-19 virus in the abnormal LTs.

### 3.8. Molecular Insights into Drugs Causing DILI in COVID-19 Patients

In COVID-19 patients with RUCAM-based DILI, various molecular mechanisms leading to the liver injury are under discussion [16,32,34,38,40,60]. As an example, there are considerations that DILI by tocilizumab (TCZ), a humanized recombinant monoclonal antibody, which functions as an IL-6 receptor antagonist fighting against the cytokine storm, may be triggered by its binding to IL-6 receptors [16]. Polypharmacy and pretreatment with potentially hepatotoxic drugs such as lopinavir/ritonavir could have facilitated the development of TCZ liver injury via molecular drug–drug interactions and drug-metabolizing enzyme induction. In a large case series, hepatic steatosis was considered as a risk factor of liver injury [32]. As this type of liver disease is closely related to overweight and obesity, both of which commonly exert induction of hepatic microsomal cytochrome P450 (CYP) 2E1, a possible role of CYP 2E1 can be assumed.

Another report focused on molecular details related to CYP 3A4, potently inhibited by ritonavir, which could promote liver injury by azithromycin through molecular mechanisms at the CYP level [34]. Molecular mechanisms for enhancing DILI during inflammation could also be associated with the production of ROS (reactive oxygen species) by inflammatory cells, possibly via myeloperoxidase, an enzyme located in inflammatory cells such as macrophages and neutrophils, in addition to immune mechanisms shown in a small subset of DILI cases [34]. For liver injury by dabigatran, the molecular mechanism of injury was assumed to be related to an idiosyncratic rather than intrinsic reaction [38]. The DILI by favipiravir or its metabolites was ascribed at the molecular level primarily to an idiosyncratic reaction [40]. However, continuous drug use causes self-inhibition of its hepatic metabolism, which may increase the favipiravir/inactive metabolite ratio, a possible contributory factor for the injury similar to a high drug intake [40]. Lastly, there is another note that a high loading dose of the drug and liver injury by the use of other hepatotoxic drugs may represent the molecular and mechanistic basis of the liver injury [60].

Although molecular aspects were discussed for some drugs involved in RUCAM-based DILI observed among the special COVID-19 cohort [16,32,34,38,40,60], providing a uniform molecular concept fitting to all drugs under consideration is not feasible due to the high number of drugs involved (Table 1). However, for single drugs such as acetaminophen, molecular steps via CYP, preferentially its isoform 2E1, leading to liver injury are commonly known and applicable to patients using this drug in overdose. Acetaminophen causes predictable intrinsic liver injury as opposed to the majority of the other drugs that cause unpredictable idiosyncratic liver injury. At a molecular level, polypharmacy, well documented in COVID-19 patients (Table 1), represents a higher risk for DILI as compared with the use of a single drug. As an example, patients were described who used 18 different drugs. It is well known that polypharmacy increases the risk of DILI, shown by th0 increased risk by a factor of six if two or more hepatotoxic drugs were used together [80].

Of course, a variety of reports exist dealing with molecular aspects of DILI, but these suggestions were not based on DILI cases assessed for causality using RUCAM. There is a report describing that the combination of lopinavir with ritonavir increases the risk of liver injury by a factor of four [5]. This suggests that one drug (lopinavir) may increase the hepatotoxic potential of the other drug (ritonavir), or vice versa. The molecular mechanism of the liver injury by hydroxychloroquine was assumed to be related to reactive metabolites and oxidative stress induced by this drug or an idiosyncratic or synergistic effect associated with inflammatory processes caused by the infection itself [4]. Several liver injury mechanisms were proposed such as oxidative stress for azithromycin, hydroxychloroquine, and lopinavir/ritonavir [11].

At the very least, clinical case analysis did not provide sufficient evidence that drug molecules may interact with liver cell subcellular structures of COVID-19 patients differently compared with not infected individuals. It is known that 62% of the drugs causing DILI are metabolized by CYP isoenzymes, and a similar percentage can be assumed for drugs used by COVID-19 patients (Table 1 and Table 2). For the COVID-19 cohort, no data on blood exosomes containing CYP isoforms are available that could identify specific isoforms implicated in DILI development. Unanswered is also the question whether hepatic hypoxia can change the molecular events leading to greater injury of CYP-dependent reactions by changing, for instance, from an oxidative pathway of the implicated drug to a more critical reductive pathway, commonly assumed in DILI caused by the anesthetic halothane. To study this question, serum LTs including glutamate dehydrogenase should be determined before and during mechanistic oxygenation of COVID-19 patients suffering under hypoxia.

### 3.9. Recommendations for Future Studies

Evaluating the role of DILI in COVID-19 patients in upcoming studies remains a clinical challenge and requires special care to obtain robust data ready to be published in qualified journals. Recommendations include the following:

(1) At admission of COVID-19 patients, LTs should be analyzed, associated with documentation of drug use prior to hospital admission and exclusion of preexisting liver diseases or acute hepatitis infections;

(2) The study protocol should be based on a prospective approach. This ensures early collection of complete case data during the hospital stay of the patient;

(3) The proactive use of the prospective RUCAM should be used in its updated version published 2016 [73] to assess causality for drugs with high RUCAM-based causality gradings, facilitated by complete datasets;

(4) The updated RUCAM should be used [73] because this version is, together with the original RUCAM published in 1993 [72], the worldwide most commonly used diagnostic algorithm to assess causality for drugs in suspected DILI and for herbs in suspected herb-induced liver injury (HILI). This was verified for 81,856 RUCAM-based DILI cases and 14,029 RUCAM-based HILI cases published from 1991 up to mid-2020 [79]. RUCAM, therefore, outnumbers any other method attempting to assess causality in suspected DILI cases [73,79]. Two types of the updated RUCAM exist, one for hepatocellular injury and one for cholestatic/mixed injury, to be assessed using ALT and ALP as parameters, as published with all details on how to manage [73]. The final study cohort should include only cases with a probable or highly probable causality level to provide homogeneity of data, facilitating analysis and a comparison of results among countries;

(5) RUCAM is based on seven domains comprising key elements that are defined and provide individual scores [73];

(6) Among the RUCAM domains are the time to onset from the beginning (or the cessation) of the drug use (scores +2 or +1), course of ALT/ALP after cessation of the drug (scores +3 to −2), risk factors (scores +1 or 0), concomitant drug(s) (scores 0 to −3), search for alternative causes (scores +2 to −3), knowledge of product hepatotoxicity (scores +2 to 0), and response to unintentional re-exposure (scores +3 to −2) [73];

(7) The score range reflects the variability of some criteria and allows for a selection of a precise attribution, avoiding a black or white choice. With a range of +14 to −9 points, the final score by drugs indicates the causality level: score ≤0, excluded causality; 1–2, unlikely; 3–5, possible; 6–8, probable; ≥9, highly probable [73];

(8) For each individual drug, a causality grading should be provided. Lumping causality levels for several drugs must be avoided for reasons of transparency and clarity;

(9) The final study cohort should include only patients with liver injury caused by drugs, thereby excluding herbs or dietary herbal products that that may cause HILI but not DILI;

(10) Liver injury criteria should prospectively be clearly defined using threshold values for ALT and ALP according to published outlines [73];

(11) With emerging new COVID-19 variants, the respective type should be searched for and presented in the final publication;

(12) Studies reporting increased LTs and drug use should clearly define the date of data acquisition: at hospital admission, during hospital stay, or after demission;

(13) The Drug-Induced Liver Injury (DILI) Study Group of the Chinese Society of Hepatology (CSH) and the Chinese Medical Association (CMA) also recommend the use of RUCAM in its updated version for assessing causality in DILI cases [81], a kind reminder not only for DILI experts in China but also worldwide.

(14) Lastly, studies are highly appreciated that correlate PO_2_ values before and during oxygen supplementation with LTs to find out whether a correlation exists in COVID-19 patients between hypoxia and abnormal LT parameters possibly related to hypoxic liver injury.

## 4. Conclusions

Pathogenetic considerations of the liver injury caused by the used drugs focus on the catalytic circle of cytochrome P450 with its isoforms and the generation of ROS in the oxidative stress setting; however, a unifying pathogenetic step applicable for all drugs cannot be retrieved from the published reports. Polymedication, a common feature in COVID-19 patients, is certainly a major risk factor of DILI. Future studies on DILI in COVID-19 require special attention by focusing on a prospective study approach and using the prospective RUCAM in its updated version to ensure complete case datasets that can help achieve a high RUCAM-based causality scoring and grading. Currently, clinical DILI characteristics in COVID-19 patients are now well defined following analysis of published RUCAM-based DILI cases among infected patients. Details of DILI features are provided for age and gender of the patients, drugs causing the injury, type of liver injury, laboratory data with maximum LT abnormalities, and outcome. Good evidence exists that DILI confounds previously published clinical features of COVID-19 patients, which now need to be redefined without DILI elements to be removed before redefinition.

## Figures and Tables

**Figure 1 ijms-23-04828-f001:**
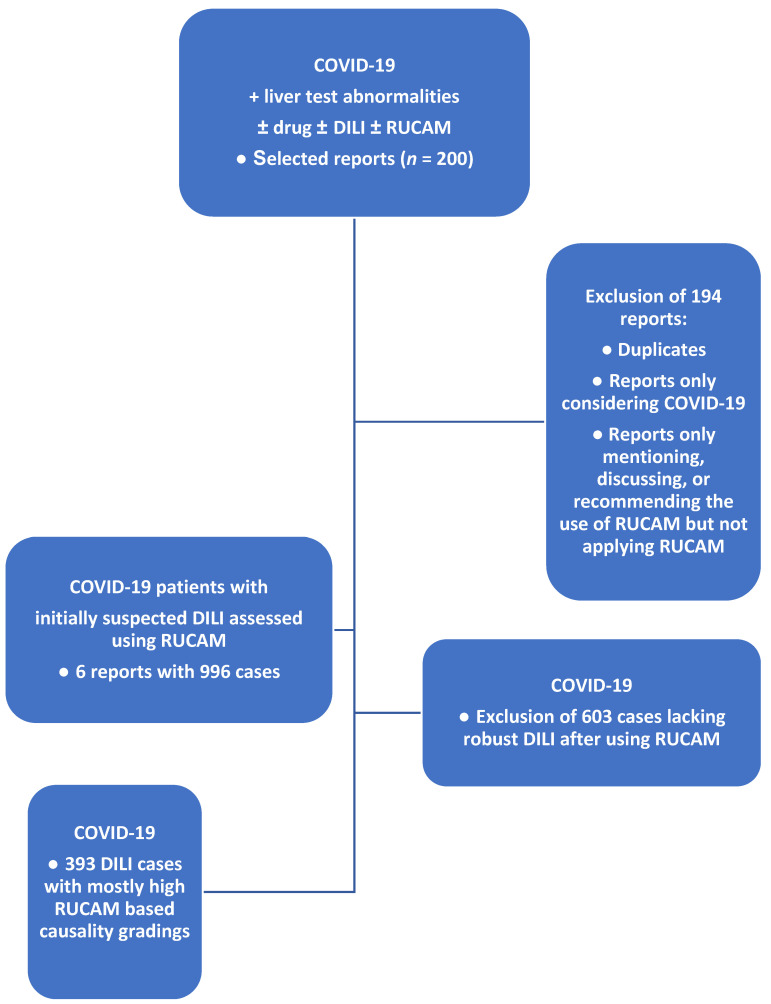
Flow diagram of literature search and study process of RUCAM-based DILI cases in COVID-19 patients. Abbreviations: COVID-19, coronavirus disease 2019; DILI, drug-induced liver injury; RUCAM, Roussel Uclaf Causality Assessment Method.

**Table 1 ijms-23-04828-t001:** COVID-19 reports published in2020 and 2021 with liver test abnormalities ± DILI ± RUCAM ± individual drugs.

First Author	COVID-19 Publications with LT Abnormalities ± DILI ± RUCAM ± Suspected Drugs (Selection)	Causality Assessment
**Publication Year: 2020**		
Bertolini [3]	Acetaminophen, lopinavir, remdesivir, ritonavir	No
Brito [4]	Azithromycin, chloroquine, ganciclovir, hydroxychloroquine, lopinavir, oseltamivir, remdesivir, ritonavir, tocilizumab	No
Cai [5]	Interferon, lopinavir, oseltamivir, ribavirin, ritona	No
Chen [6]	Ganciclovir, linezolid, lopinavir, oseltamivir, ritonavir, tigecycline	No
Chen [7]	Oseltamivir	No
Chu [8]	Abidol, interferon, litonavir, lopinavir, ribavirin	No
Da Silva [9]	Drugs not specified	Updated RUCAM recommended
Feng [10]	Abidol, acetaminophen, azithromycin, ceftazidime, levofloxacin, linezolid, lopinavir, meropenem, oseltamivir, paramivir	No
Ferron [11]	Azithromycin, hydroxychloroquine, lopinavir, remdesivir, ritonavir, tocilizumab	Use of RUCAM encouraged
Grein [12]	Remdesivir	No
Kulkarni [13]	Abidol, acyclovir, azithromycin, chloroquine, darunavir, levofloxacin, lopinavir, oseltamivir, remdesivir, ritonavir, umifenovir	No
Méndez- Sánchez [14]	Acetaminophen, lopinavir, oseltamivir, ritonavir	No
Meszaros [15]	Amoxicillin-clavulanic acid, azithromycin, hydroxychloroquine, lopinavir, remdesivir, ritonavir	No
Muhović [16]	Azithromycin, ceftriaxone, methylprednisolone, tocilizumab	RUCAM used
Nardo [17]	Acetaminophen, hydroxychloroquine, lopinavir, remdesivir, ritonavir, tocilizumab	No
Olry [18]	Azithromycin, baricitinib, favipiravir, hydroxychloroquine, imatinib, interferon, lopinavir, remdesivir, ritonavir	No
Serviddio [19]	Azithromycin, hydroxychloroquine, lopinavir, ritonavir, tocilizumab	No
Wang [20]	Baricitinib, interferon, lopinavir, remdesivir, ribavirin, ritonavir, tocilizumab	No
Weber [21]	Acetaminophen, azithromycin, interferon, lopinavir, piperacillin-tazobactam, ritonavir	No
Wu [22]	Arbidol, acetaminophen, hydroxychloroquine, lopinavir, ritonavir	No
Xu [23]	Ritonavir, lopinavir, ribavirin	No
Xu [24]	Ritonavir, lopinavir	No
Xu [25]	Abidol, interferon, lopinavir, ritonavir	No
Yoshida [26]	Drugs not specified	No
Zhang [27]	Drugs not specified	No
**Publication Year: 2021**		
Afify [28]	Acetaminophen, azithromycin, chloroquine, colchicine, lopinavir, remdesivir, ritonavir, tocilizumab, umifenovir	No
Badedi [29]	Azithromycin, favipiravir, lopinavir, remdesivir, ribavirin, ritonavir	No
Bloom [30]	Acetaminophen, azithromycin, hydroxychloroquine, lopinavir, methylprednisolone, remdesivir, ritonavir, tocilizumab	No
Boettler [31]	Anakinra, arbidol, azithromycin, baricitinib, camostat, chloroquine, emapalumab, favipiravir, hydroxychloroquine, lopinavir, methylprednisolone, remdesivir, ribavirin, ritonavir, sofobuvir, tocilizumab	No
Chen [32]	Abidol, acetaminophen, oseltamivir, ribavirin	Updated RUCAM used
Clinton [33]	Lopinavir, remdesivir, ritonavir, tocilizumab	Updated RUCAM recommended
Delgado [34]	Acetaminophen, azithromycin, ceftriaxone, dexketoprofen, doxycycline, enoxaparin, hydroxychloroquine, interferon, levofloxacin, lopinavir, metamizole, omeprazole, pantoprazole, piperacillin/tazobactam, remdesivir, ritonavir, tocilizumab	Updated RUCAM used
Gaspar [35]	Lopinavir, remdesivir, ritonavir, tocilizumab	No
Huang [36]	Abidol, acetaminophen, chloroquine, interferon, lopinavir, remdesivir, ritonavir, tocilizumab	No
Jiang [37]	Abidol, amoxicillin, ceftezole, cetirizine, flucloxacillin, interferon, lopinavir, oseltamivir	No
Jothimani [38]	Enoxaprin, esomeprazole, dabigatran, methylprednisolone	RUCAM used
Kalal [39]	Amoxicillin–clavunate, azithromycin, ceftriaxone, hydroxychloroquine, meropenem, piperacillin, tazobactam, tocilizumab	No
Kumar [40]	Acetaminophen, enterkavir, favipiravir	RUCAM used
Lin [41]	Abidol, interferon, lopinavir, ritonavir	No
Marjot [42]	Lopinavir, remdesivir, ritonavir, tocilizumab	No
McGrowder [43]	Acetaminophen, azithromycin, hydroxychloroquine, lopinavir, remdesivir, ritonavir, tocilizumab	No
Moreira [44]	Acetaminophen, hydroxychloroquine, lopinavir, remdesivir, ritonavir, tocilizumab	No
Omar [45]	Acetaminophen, favipiravir, hydroxychloroquine, interferon, lopinavir, oseltamivir, remdesivir, ritonavir, tocilizumab	No
Ortiz [46]	Azithromycin, hydroxychloroquine, ivermectin, lopinavir, nevirapine, ritonavir, remdesivir, tocilizumab	Use of RUCAM recommended
Satsangi [47]	Azithromycin, favipiravir, hydroxychloroquine, ivermectin, lopinavir, remdesivir, ritonavir, tocilizumab	No
Sharma [48]	Drugs not specified	No
Sharma [49]	Azathioprine, azithromycin, chloroquine, hydroxychloroquine, interferon, lopinavir, ritonavir, remdesivir, tocilizumab	No
Shousha [50]	Azithromycin, chloroquine, hydroxychloroquine	No
Sivandzadeh [51]	Abidol, acetaminophen, favipiravir, lopinavir, remdesivir, ritonavir	No
Sodeifian [52]	Azithromycin, favipiravir, hydroxychloroquine, interferon, lopinavir, oseltamivir, remdesivir, ribavirin, ritonavir, tocilizumab	Use of RUCAM recommended
Teschke [53]	Drugs not specified	Updated RUCAM recommended
Vidal-Cevallos [54]	Drugs not specified	Updated RUCAM recommended
Vitiello [55]	Acetaminophen, lopinavir, remdesivir, ritonavir	No
Wang [56]	Abidol, acetaminophen, baricitinib, interferon, lopinavir, methylprednisolone, ribavirin, ritonavir, tocilizumab	No
Weber [57]	Drugs not specified	No
Wu [58]	Acetaminophen, chloroquine, hydroxychloroquine, interferon, lopinavir, piperacillin/tazobactam, remdesivir, ribavirin, ritonavir, tocilizumab	Updated RUCAM recommended
Yadav [59]	Lopinavir, ritonavir	No
Yamazaki [60]	Favipiravir, interferon-β, lopinavir, meropenem, micafungin, ritonavir, trimethoprim–sulfamethoxazole, vancomycin	Updated RUCAM used
Yip [61]	Interferon-β, lopinavir, hydrocortisone, methylprednisolone, oseltamivir, ribavirin, ritonavir	No
Zhang [62]	Abidol, chloroquine, lopinavir, oseltamivir, ribavirin, ritonavir	No

Listed are selected drugs specifically used by patients with COVID-19 infections. Most of the patients also received other, not further identified drugs from the groups of antibiotics, antivirals, antifungals, and glucocorticosteroids. RUCAM refers to the original RUCAM published in 1993, whereas the updated RUCAM was published in 2016. Abbreviations: COVID-19, coronavirus disease 2019; DILI, drug-induced liver injury; RUCAM, Roussel Uclaf Causality Assessment Method.

**Table 2 ijms-23-04828-t002:** Reports of COVID-19 patients with DILI assessed for causality using RUCAM.

First Author Country Cases (*n*) Drugs (*n*)	Case Details of RUCAM Based DILI in COVID-19 Patients
Muhović, 2020 [16] Montenegro (cases, *n* = 1) (drugs, *n* = 4)	Reported is a case of DILI by tocilizumab (TCZ) in a male patient with COVID-19 infection that caused a cytokine storm [16]. Using the original RUCAM [72] instead of the commonly preferred updated RUCAM [73], causality for TCZ was probable according to a RUCAM score of 8. Such high causality grading is commonly achieved with complete datasets asked for prospective use during the clinical course. TCZ is a humanized recombinant monoclonal antibody that acts as an IL-6 receptor antagonist by specific binding to IL-6 receptors. Preexisting liver disease was excluded, as well as anoxia leading to liver hypoxia. It was noted that slightly elevated transaminases were detected before TCZ hepatotoxicity was observed, conditions seen in other COVID-19 patients with a severe clinical course. Co-medication included azithromycin, ceftriaxone, chloroquine, lopinavir, methylprednisolone, and ritonavir, but none of these drugs were considered causative for the liver injury, although a contributory role of the previously used antiviral drugs (lopinavir/ritonavir) is possible.
Chen, 2021 [32] China (cases, *n* = 830) (discussed drugs, *n* = 4)	Analyzed were 830 COVID-19 cases with liver injury. This is the largest study cohort evaluated for causality [32] using the updated RUCAM [73]. Among 74/830 cases, the RUCAM score was >3, corresponding to a possible, probable, or highly probable causality grading. To achieve a homogeneous cohort, a good approach would have been to include only cases with a probable or highly probable causality ranking. Discussed were the drugs abidol, acetaminophen, oseltamivir, and ribavirin. For this retrospective study, all data were retrieved from the digital medical records during hospitalization. As a specific appeal, when multiple drugs in combination are used in COVID-19 patients, the RUCAM score is required to evaluate the risk of DILI.
Delgado, 2021 [34] Spain (cases, *n* = 160) (drugs, *n* = 18)	The updated RUCAM [73] was used in 124 males and 36 female patients [34], providing in 82/160 patients a probable causality grading according to a RUCAM score of ≥6 and in 78/160 cases a possible causality ranking according to a RUCAM score of ≥3. The high possible causality grading could have been avoided by using a prospective study design. DILI was defined with ALT five times the ULN. During the hospital stay, the mean number of used drugs per patient was 14.7 (SD 7.6), whereby 98.1% received a polypharmacy with >5 drugs. Among the used drugs were acetaminophen, azithromycin, ceftriaxone, dexketoprofen, doxycycline, enoxaparin, hydroxychloroquine, interferon, levofloxacin, lopinavir, metamizole, omeprazole, pantoprazole, piperacillin/tazobactam, remdesivir, ritonavir, and tocilizumab.
Jothimani, 2021 [38] India (cases, *n* = 1) (drugs, *n* = 4)	RUCAM was used without clear definition of the version applied [72,73] in this male COVID-19 patient [38], who suffered from DILI after using the oral anticoagulant dabigatran, for which a RUCAM score of 7 corresponding to a probable causality grading was verified. Additional medications included enoxaparin, esomeprazole, and methylprednisolone. It was outlined that the cause of liver injury is multifactorial in COVID-19, with difficulty pinpointing the exact cause.
Kumar, 2021 [40] India (cases, *n* = 3) (drugs, *n* = 3)	In this study of three patients (two females, one male) with COVID-19, each treated with favipiravir that caused DILI, RUCAM was used without specifying the RUCAM version applied [40]. The updated RUCAM was likely used, which requires the exclusion of hepatitis E virus (HEV) infection [73], a parameter considered in the present study [40] that was not an element of the original RUCAM [72]. For all three patients, a RUCAM score of 7 was presented, consistent with a probable causality level [40]. Of note, the second patient also used acetaminophen, and the third patient was also under treatment with entecavir for his hepatitis B-related cirrhosis, currently with a negative hepatitis B DNA titer.
Yamazaki, 2021 [60] Japan (cases, *n* = 1) (drugs, *n* = 8)	The updated RUCAM [73] was used in a male COVID-19 patient experiencing DILI by favipiravir, providing a RUCAM score of 6 in line with a probable causality grading and not a possible level as erroneously published [60]. The patient received a multiple medications that included interferon-β, lopinavir, meropenem, micafungin, ritonavir, trimethoprim–sulfamethoxazole, and vancomycin. A contributory role of vancomycin and meropenem was discussed.

Abbreviations: COVID-19, coronavirus disease 2019; DILI, drug-induced liver injury; RUCAM, Roussel Uclaf Causality Assessment Method.

**Table 3 ijms-23-04828-t003:** Clinical characteristics of RUCAM-based DILI in 393 COVID-19 patients.

Basic Clinical Characteristics of RUCAM-Based DILI in 393 COVID-19 Patients	Cohort Cases (*n*)	References
**Gender**		
Male, 1 case	1	Muhović [16]
Males–females = 145:82 = 1.8:1.0, 227 cases	259	Chen [32]
Males–females = 124:36 = 3.4:1.0, 160 cases	160	Delgado [34]
Male, 1 case	1	Jothimani [38]
Males–females = 1:2 = 0.5:1.0, 3 cases	3	Kumar [40]
Male	1	Yamazaki [60]
**Age**		
52 years, 1 case	1	Muhović [16]
56 (median) years, 227 cases	259	Chen [32]
54.3 (mean) years, 160 cases	160	Delgado [34]
51 years, 1 case	1	Jothimani [38]
57.3 years (mean), 3 cases	3	Kumar [40]
73 years, 1 case, 1 case	1	Yamazaki [60]
**RUCAM-based causality grading**		
Score 8, probable, 1 case	1	Muhović [16]
Score >3, possible, probable, highly probable, 227 cases	259	Chen [32]
Score 6–8, probable, 82 cases	160	Delgado [34]
Score 3–5, possible, 78 cases	160	Delgado [34]
Score 7, probable, 1 case	1	Jothimani [38]
Score 7, probable, 3 cases	3	Kumar [40]
Score 6, probable, 1 case	1	Yamazaki [60]
**Laboratory data**		
ALT, U/L		
1541, 1 case	1	Muhović [16]
135.8 (median), 32 cases	259	Chen [32]
465.8 (mean), 160 cases	160	Delgado [34]
41, 1 case	1	Jothimani [38]
906.7 (mean), 3 cases	3	Kumar [40]
115, 1 case	1	Yamazaki [60]
AST, U/L		
1076, 1 case	1	Muhović [16]
78.9 (median), 32 cases	259	Chen [32]
N.A.	160	Delgado [34]
36, 1 case	1	Jothimani [38]
640.3 (mean), 3 cases	3	Kumar [40]
268, 1 case	1	Yamazaki [60]
ALT/AST		
1.43, 1 case	1	Muhović [16]
0.84, 32 cases	259	Chen [32]
N.A.	160	Delgado [34]
1.08, 1 case	1	Jothimani [38]
1.42	3	Kumar [40]
0.43	1	Yamazaki [60]
ALP, U/L		
Normal, 1 case	1	Muhović [16]
N.A.	259	Chen [32]
150.7 (median), 160 cases	160	Delgado [34]
298, 1 case	1	Jothimani [38]
590.7 (mean), 3 cases	3	Kumar [40]
710, 1 case	1	Yamazaki [60]
**Liver injury type**		
Hepatocellular		
1 case	1	Muhović [16]
N.A.	259	Chen [32]
92 cases	160	Delgado [34]
3 cases	3	Kumar [40]
Cholestatic		
N.A.	259	Chen [32]
6 cases	160	Delgado [34]
1 case	1	Jothimani [38]
1 case	1	Yamazaki [60]
Mixed		
20 cases	160	Delgado [34]
Not classified		
42 cases	160	Delgado [34]
**Clinical outcome**		
Recovery		
1 case	1	Muhović [16]
141 cases	160	Delgado [34]
Death	160	Delgado [34]
1 case associated with the disease		
18 cases unrelated with the disease	160	Delgado [34]

Abbreviations: COVID-19, coronavirus disease 2019; DILI, drug-induced liver injury; RUCAM, Roussel Uclaf Causality Assessment Method.

## Data Availability

Data are available from the published reports as referenced in detail.

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
