# Peer review of "Liver Injury in COVID-19 Patients with Drugs as Causatives: A Systematic Review of 996 DILI Cases Published 2020/2021 Based on RUCAM as Causality Assessment Method"

_ijms, 2022, doi:10.3390/ijms23094828_

Round 1

Reviewer 1 Report

This review of  liver parenhimal damage  which s manily manifested by an increase in liver transaminases (LTs). LTs can also be caused by SARS Covid 2 virus as well as other causes that have been analyzed in detail and listed in the article. Occasionally,it is difficult to discern the causes of LTs increase, and wheather LTs increase consequence of COVID infection or other drugs, or allternative therapies that patients  very often take, especially in Asian countries.

I suggest to take a look at the references.

Author Response

Answer for reviewer 1:

Thank you for your valuable comments. We included a clarifying statement under section 3.5, last sentence shown in red: Of note, RUCAM based case evaluation was done by the quoted authors and not redone by us as case details were not available to us but confined to the authors of their published work. The use of RUCAM allowed them to exclude alternative causes to be differentiated from DILI.

It is expected that the authors of the original papers did exclude potential alternative causes.

Reviewer 2 Report

The paper « Liver injury in COVID-19 patients with drugs as causatives was written by Teschke et al. who are experts in the field of Drug Induced Liver Injury.

The paper try to summarize the litterature on DILI in patients treated for COVID-19.

The authors show that many different mechanisms can be involved in liver injury induced by COVID : inflammation, hypoxia, direct toxicity of the virus and DILI. As many patients are obese, NAFLD can also explain some of the changes of the liver function tests.

They included six papers to study the drugs involved and the causal relationship with liver injury in patients with COVID.

All the studies included are retrospective. The RUCAM threshold used to suggest a causal relationship was different between the studies. Is it reasonable to include the study of Chen et al. ?

Four papers included are case reports. Then, only the study of Delgado et al. includes enough patients.

The authors use the RUCAM 2016 » score as the « alpha and omega « to study the causal relationship between the drugs and liver injury.

Generally speaking, I think that we should be very humble when we describe the tools we use to study the causal relationship between a drug and liver injury.

The authors should emphasize the limits of the RUCAM score.

For example :

-what is the evidence that age and alcohol are risk factors for the drugs used in patients with COVID ?

-what is the evidence to use ALT > 5 ULN to define cytolytic hepatitis ? In the context of COVID should we use 8N ? Of course if bili stays < 2N.

-should we use the same criteria in patients with obesity and/or steatosis (Chalasani et al. AJG 2021) ?

-Is liver toxicity the same in patients with obesity with and without NAFLD : Fromenty et al. have published several papers on this topic. Most of the studies were done on animals and we don’t know if the same mechanisms are involved in humans (Massart et al. IJMS 2022). A recent paper from Korea support this idea in patients with NAFLD  but the risk factors are not known (Hwang et al. CGEH 2022) ?

-Will the toxicity will be the same in a patient with and without inflammation ?

-If the patient has NAFLD and inflammation, is RUCAM the good tool ?

-When you have 3 drugs with a RUCAM score > 6, do you stop all the 3 drugs ?

-How many prospective studies where done to validate the RUCAM score.

I am a general hepatologist with 30 years of experience. I think that DILI is one of the most difficult topics in hepatology.

In summary I think that the authors should be more balanced.

Author Response

Answer to reviewer 2:

Thank you for providing constructive comments and appreciating our expertise in DILI.

  1. Yes, obesity is a common feature in COVID-19 patients according to literature, but this was not discussed or quantified in the 6 RUCAM based DILI papers quoted by us. Prospective studies in a larger population are lacking to clearly show that metabolic associated fatty liver disease (MAFLD) may contribute to increased LTs in DILI; as this topic is still highly speculative, we prefer not to include this kind of speculation. In addition and as compared with DILI, increases of LTs in MAFLD are much smaller and below ALT threshold values required for DILI.
  2. We prefer to keep the paper of Chen, to provide a balanced and broader view.
  3. Authors of the DILI patients with COVID-19 used RUCAM without any criticism, in line with 81,856 published DILI cases worldwide (our ref. 79). Discussion of limitations, risk factors (alcohol and age), and thresholds of RUCAM is outside the scope of this systematic review article.
  4. Steatosis an inflammation: Liver histology is not an element of RUCAM, therefore data are lacking and cannot be discussed.
  5. Three drugs with a RUCAM score >6: one case may have a score of 7, another one of 8, and a third one of 9, thereby showing different levels of causality gradings. If all drugs have, for instance, a score of 9, we would stop all 3 drugs.
  6. With 50 years of experience, I (RT) agree that DILI is one of the most difficult topics in hepatology, but RUCAM helps solve problems.

Reviewer 3 Report

The authors conducted a review of the literatures regarding liver injury and especially DILI based on the RUCAM in COVID-19 patients.
The suspect drugs and characteristics of DILI in COVID-19 are summarized in Table 1-3, and they will be of interest for the readers.

I feel that some parts of the manuscript could be improved before publication.

Please check whether the following sentence is grammatically correct.
(Line 17-19 in Abstract) 
Based on finally 6/200 reports with initially suspected 996 DILI cases in COVID-19 patients and using all RUCAM based DILI cases allowed for a clear description of clinical features of on RUCAM based DILI cases were sparseDILI in COVID-19 patients.

Please make it more clear how the authors selected 200 articles (line 85).
For example, I could not find any article in PubMed database (https://pubmed.ncbi.nlm.nih.gov/) using the combination of the five terms "COVID-19, liver test abnormalities, drug, DILI, RUCAM".
I found only 14 articles, which were much fewer than the 3,000 hits (line 70-71), in PubMed Central database using the same combination (https://www.ncbi.nlm.nih.gov/pmc/?term=COVID-19%2C+liver+test+abnormalities%2C+drug%2C+DILI%2C+RUCAM). 
Moreover, how did the authors identified the 60 articles listed in Table 1?

Please align rows in each column in Table 1. It is hard to read.

Author Response

Answer to reviewer 3:

Thank you for your valuable comments and mentioning the possible interest for the readers.

  1. Correction in abstract was done in red.
  2. We used also google and corrected it and provide details in red under section 2.
  3. Formatting of Table 1 was improved, thank you for suggestion.

Round 2

Reviewer 2 Report

1) I am not sure that the authors have understood my previous comment.

In a situation like COVID, I am not sure that RUCAM can be applied without caution.

Example : if the patient has NAFLD with ALT = 3N at baseline and is treatd with a drug and his ALT increases to 6N.

We can speculate that the drug is responssible of the toxicity. I am not sure that in this situation we have to stop the drug and I am not sure of the opposite.

Do I have to stop this drug ? What can I do if the drug is the only drug that is efficient.

There are so many factors involved

2) I think that the authors should discuss the limits of the RUCAM score in a situation where first there are many different etiologies of liver injury and second the different etiologies can interact

Author Response

Response for reviewer 2

Thank you for your patience, efforts to improve, and valuable comments, all were considered in a new para on P10 shown in red:

  1. A cautionary is now included. NAFLD/MAFLD is now discussed. Stopping drug is now mentioned and discussed. Considering factors is now done.
  2. Limits of RUCAM elements and scores are now discussed. Interactions are now discussed.

Round 3

Reviewer 2 Report

Thank you for your response